# Development of a Pentacistronic Ebola Virus Minigenome System

**DOI:** 10.3390/v17050688

**Published:** 2025-05-09

**Authors:** Brady N. Zell, Vaille A. Swenson, Shao-Chia Lu, Lin Wang, Michael A. Barry, Hideki Ebihara, Satoko Yamaoka

**Affiliations:** 1Virology and Gene Therapy Program, Mayo Clinic Graduate School of Biomedical Sciences, Rochester, MN 55905, USA; zell.brady@mayo.edu (B.N.Z.); swenson.vaille@mayo.edu (V.A.S.); 2Division of Infectious Diseases, Department of Medicine, Mayo Clinic, Rochester, MN 55905, USA; lu.shao-chia@mayo.edu (S.-C.L.); wang.lin@mayo.edu (L.W.); 3Department of Immunology, Mayo Clinic, Rochester, MN 55905, USA; 4Department of Virology 1, National Institute of Infectious Diseases, Musashimurayama 208-0011, Japan; hebihara@niid.go.jp

**Keywords:** Ebola virus, minigenome, trVLPs, mouse-adaptation

## Abstract

Ebola virus (EBOV) causes severe disease outbreaks in humans with high case fatality rates. EBOV requires adaptation to cause lethal disease in mice by acquiring single mutations in both the nucleoprotein (NP) and VP24 genes. As an attempt to model mouse-adapted EBOV (MA-EBOV), we engineered novel pentacistronic minigenomes (5xMG) containing a reporter gene, VP40, and glycoprotein genes as well as the NP and VP24 genes from either EBOV or MA-EBOV. The 5xMGs were constructed and optimized, and the produced transcription- and replication-competent virus-like particles (trVLPs) were demonstrated to infect several cell lines. Introduction of the mouse-adaptation mutations did not significantly impact the replication and transcription of the 5xMG or the relative infectivity of the trVLPs in vitro. This work demonstrates the development of the 5xMG system as a new versatile tool to study EBOV biology.

## 1. Introduction

Ebola virus (EBOV) has caused multiple outbreaks of EBOV disease (EVD) since the late 20th century [1]. With high case fatality rates and few approved therapeutic options, EVD continues to pose a significant threat to global health and has been classified as a priority disease by the World Health Organization (WHO) [2,3]. Due to its extreme virulence, infectious EBOV must be handled in biosafety level 4 (BSL-4) laboratories, significantly limiting the breadth of molecular research into EBOV biology and pathogenesis.

EBOV has a single-stranded, negative-sense RNA genome, which encodes seven structural genes [4]. The genome is encapsidated in a nucleocapsid and enclosed within a host-derived lipid bilayer membrane enveloped particle. Life cycle modeling systems, like minigenomes, are powerful tools for studying the molecular biology of EBOV without the safety concerns associated with handling infectious virus (reviewed in [5]). Minigenomes are truncated, defective viral genomes lacking several or all viral open reading frame (ORF) sequences, which are replaced by reporter genes to monitor their activity in cells. Importantly, minigenomes retain non-coding regions containing the minimal promoter recognized by the viral polymerase as well as the genome packaging signals. This allows minigenomes to undergo replication and transcription when viral ribonucleoprotein (RNP) complex proteins, including nucleoprotein (NP), transcriptional activator VP30, polymerase cofactor VP35, and RNA-dependent RNA-polymerase (L), are provided in trans [6]. Additionally, by supplying additional viral structural proteins, such as matrix protein VP40, surface glycoprotein (GP), and VP24, the minigenome systems can be expanded to the production of transcription- and replication-competent virus-like particles (trVLPs) [7,8]. trVLPs are morphologically similar to EBOV particles and have been used for studying EBOV biology as well as for evaluation as a vaccine candidate [8,9,10,11].

There are well-established animal models for EVD in non-human primates, pigs, ferrets, hamsters, guinea pigs, and mice (reviewed in [12]). While mice can be used, it is worth noting that EBOV does not cause significant diseases in most naïve mice. However, EBOV can be adapted to cause lethal disease in mice through serial passaging in progressively older suckling mice [13,14,15]. After this selection, mouse-adapted Ebola virus (MA-EBOV) has been shown to recapitulate key hallmarks of human EVD, including high viremia, extensive pro-inflammatory cytokine activation [13,16], and lymphocyte apoptosis [17,18,19,20].

When the sequence of MA-EBOV was examined, 13 nucleotide changes were acquired. Five mutations did not change amino acid codons, but the eight others mutated amino acids [13,14,15]. When the single amino acid changes were examined in mice, the single mutations in NP and VP24 were critical for a virulent phenotype only when both were present in the virus [21]. These mutations have been linked to overcoming host type I interferon (IFN) response and enhancing viral replication in mouse macrophages [21]. However, the detailed roles of these mutations still remain unclear.

In this study, we established a pentacistronic EBOV minigenome (5xMG) by introducing the NP gene, expanding on the previously established minigenome systems already containing VP24, VP40, GP, and a reporter gene. To create a system to evaluate the effects of mutations found in MA-EBOV, we introduced the mutations into the NP and VP24 genes within the 5xMG and demonstrated the ability to produce infectious trVLPs. We propose the 5xMG as a new tool for the assessment of EBOV biology under BSL-2 conditions.

## 2. Results

### 2.1. Development of an NP Gene-Containing Pentacistronic EBOV Minigenome

A previously described tetracistronic minigenome (4xMG) expresses VP40, GP, VP24, and a luciferase reporter gene [7]. With the intent to model MA-EBOV, we expanded the system to a 5xMG system by incorporating the EBOV NP gene between the NP untranslated region (UTR) and VP35 transcription start signal, its “native” position in the EBOV viral genome (Appendix A). This construct, flanked by a hammerhead ribozyme and hepatitis delta virus ribozyme sequences, was inserted in the antisense orientation into an expression plasmid under the control of the cytomegalovirus (CMV) promoter. The resulting 5xMG contains the viral genes NP, VP40, GP, and VP24, along with a NanoLuc luciferase reporter gene (Figure 1A).

When 5xMG is co-transfected in P0 cells with NP, VP30, VP35, and L plasmids, its genome undergoes replication and transcription, like the 4xMG, as well as assembles trVLPs that can infect P1 target cells (Figure 1B). The 5xMG delivered via trVLP infection only undergoes primary transcription in naïve P1 cells. If these cells are previously transfected with the VP30, VP35, and L plasmids, the 5xMG genome is replicated, thereby amplifying both mRNA and protein production from the minigenome. In contrast, the 4xMG does not replicate under the same conditions as its genome does not carry the NP gene (Appendix A). The NP protein must be externally supplied to P1 cells to have 4xMG to replicate. The presence of the NP gene within the 5xMG genome should allow aspects of EBOV NP biology to be examined that are not feasible in the existing 4xMG system due to a more authentic manner of expression.

To directly compare the 5x- with 4xMG, 293T cells were transfected using previously published minigenome and helper plasmid (NP, VP35, VP30, and L) amounts and protocol [22]. Both systems showed a significant increase in reporter signal in the presence of the L polymerase (L+) when compared to an L-absent (L−) control. However, the 5xMG construct exhibited an approximately 10-fold lower relative reporter activity compared to 4xMG (Figure 2A). This aligns with previous findings that minigenome reporter activity is inversely correlated with its genome length [7], and may also be influenced by the reporter gene’s placement as the second gene within the minigenome, which will be transcribed less due to the known gradient of transcription [23].

### 2.2. Plasmid Optimization and the T7 Bacteriophage Promoter Increase 5xMG Efficiency in Producer Cells

To maximize the activity of 5xMG, we individually titrated the minigenome and helper plasmids while keeping the other plasmid amounts constant during the transfection of producer cells (Figure 2B). The initial transfection ratios were based on the previously published 4xMG system [7,22], which used 125 ng NP, 125 ng VP35, 75 ng VP30, 1000 ng L, and 250 ng minigenome plasmids in a 12-well plate format. Starting from zero, the plasmid quantity used in transfection was gradually increased up to 500–1500 ng, depending on the plasmid size. The optimized plasmid quantities, which produced the highest signal-to-noise ratio for the 5xMG, were: 75 ng NP, 125 ng VP35, 250 ng VP30, 500 ng L, and 125 ng minigenome plasmids in a 12-well plate. Compared to the starting conditions, the optimized values featured a reduction in the amounts of NP, L, and minigenome plasmids, and an increase in VP30 plasmid. Importantly, the unoptimized reactions did not reach a statistical significance between the L− and L+ conditions, while the optimized reactions showed both an increased reporter signal and a significant difference between L− and L+ (Figure 2C). All subsequent 5xMG transfections and trVLP production utilized the optimal plasmid quantities for maximum yield.

A previous study demonstrated that the CAG promoter (CAGp), an RNA polymerase II (pol-II) promoter combining the CMV enhancer with the chicken beta-actin promoter and intron, drives stronger EBOV minigenome activity compared to the CMV promoter [24]. The T7 bacteriophage polymerase promoter (T7p) has also been successfully utilized in EBOV minigenome systems [6,7,24,25,26]. To compare promoter efficiency for producing 5xMG RNA genomes, we generated two additional 5xMG constructs: one driven by CAGp and the other by T7p (Figure 2D) and assessed their minigenome activities alongside the CMVp-driven construct. Reporter assays demonstrated that each of the constructs mediated significant increases in luciferase activity when L was provided compared to the L− condition (Figure 2E). CAGp was stronger than CMVp, but T7p outperformed both, showing a 400- to 500-fold increase over CAGp and CMVp (Figure 2F). Notably, similar trends were also observed in the EBOV 4xMG system at both early and later timepoints following transfection (Appendix A, Figure 2G), suggesting that the T7p system may be more efficient than pol II-based systems in driving multi-cistronic EBOV minigenome activity.

### 2.3. 5xMG Replication Generates trVLPs Containing Detectable GP and VP40

An advantage of expressing VP40, GP, and VP24 from the minigenome is the ability to produce trVLPs from P0 producer cells that can be used to infect p1 target cells (Figure 1B). 5xMG trVLP particle integrity was assessed by a protease protection assay [27,28]. trVLPs were concentrated and separated into equal fractions before protease or detergent treatment, followed by viral protein detection by western blot. Under these conditions, VP40 protein was present in trVLPs in the absence of detergent and protease (Figure 3A, Lane 4). Treating with trypsin alone was insufficient to degrade VP40 (Figure 3A, Lane 2). If detergent was added to the particles, VP40 was digested by protease, indicating that VP40 was protected within lipid bilayer trVLPs, maintaining particle integrity. In contrast, the GP was completely digested by trypsin without the need for detergent, indicating that the glycoprotein was displayed on the surfaces of the trVLPs (Figure 3A, Lane 2). These data suggest that the 5xMG can generate production of VP40- and GP-containing trVLPs.

### 2.4. NP Is Expressed from the Pentacistronic EBOV Minigenome in trVLP-Infected Cells

The previous studies have shown that minigenomes delivered via trVLP infection undergo primary transcription in infected cells, while genome replication and further trVLP formation occur when helper proteins are supplied in trans [7]. To assess the infectivity of 5xMG trVLPs, we used the Huh7 human hepatocellular carcinoma cell line in the presence and absence of helper proteins. When none of the helper proteins were present, there was a 4.8-fold increase in reporter signal from 1 to 72 h post-infection (Figure 3B, black line). When all helper proteins were supplied to infected cells via expression plasmid transfection, the 5xMG signal was amplified over 300-fold throughout a 72-h period (Figure 3B, blue line). This robust increase in 5xMG activity in the presence of all helper proteins was consistent with previously reported findings using 4xMG [7]. Notably, if the NP helper plasmid was absent from the transfection, there was still an 87-fold increase in luciferase signal, significantly stronger than without any helpers (Figure 3B, red line). These data suggest that the NP provided from the 5xMG is sufficient to support transcription and replication of the minigenome RNA if other EBOV helper proteins are provided from plasmids. These results also reveal a 3- to 4-fold reporter signal increase when NP is transfected compared to the NP-absent condition. This could be due to the excess NP allowing for more transcription of the present minigenome RNA copies, whereas the NP-absent condition relies first on NP to be produced from the minigenome itself. The levels of NP and VP24 proteins in cell lysates lacking NP pre-transfection correlated with the increasing reporter signal (Figure 3C). When trVLP-supernatants were transferred to helper-expressing Huh7 cells over three passages, we observed a continually stable reporter signal in the target cell lysates (Figure 3D). Together, these findings indicate that 5xMG can generate infectious trVLPs and enable NP and VP24 co-expression from the minigenome in infected cells.

### 2.5. Introduction of Mouse-Adaptation Mutations Does Not Significantly Impact 5xMG Reporter Activity and trVLP Infectivity

EBOV 5xMG was next modified with the two key virulence determinant mutations in MA-EBOV: S72G in NP and T50I in VP24 (Figure 4A). MA-5xMG mediated similar activity to WT-5xMG with significant induction of luciferase when comparing L+ to L− samples in transfected 293T producer cells (Figure 4B). It is important to note that the NP helper plasmids used for transfection were matched to either WT or MA sequences for their respective minigenomes. These results indicate that the MA mutations did not appreciably impact 5xMG replication or transcription in 293T cells.

As demonstrated above with the WT-5xMG, we next tested the infectivity of the MA-5xMG trVLPs. Helper-expressing Huh7 cells were inoculated with MA-5xMG trVLPs, and cell lysates were harvested over a 72-h timeframe for luciferase reporter assays. Between 1 and 72 h post-infection, a 139-fold increase in reporter signal was observed, demonstrating the infectivity of the MA-trVLPs and amplification of their genomes.

We also transferred WT- and MA-trVLPs to a neutral cell line (i.e., not human or mouse) at different volumes to observe the relative infectivity of the samples (Figure 4D). VeroE6 cells, which are commonly used for rescue and titration of infectious WT- and MA-EBOV [29,30], were pre-transfected with helper plasmids and infected with varying dilutions of trVLPs. Downstream luciferase reporter assays revealed no statistically significant change in reporter signal from MA- compared to WT-trVLPs at 72 h post-infection (Figure 4D). Taken together, these results demonstrate that the introduction of the mouse-adaptation mutations does not make significant alterations in the capability of 5xMG replication and transcription or trVLP infectivity.

## 3. Discussion

In this work, we developed a novel minigenome system to model EBOV infections under BSL-2 conditions. Here, we generated minigenomes that produce viral RNA genomes that not only encode the standard three EBOV genes found in 4xMGs (VP24, VP40, and GP) but also NP, which plays critical roles in the viral life cycle. These 5xMGs, therefore, allow exploration of the biology of NP with MGs and trVLPs at BSL-2 that was not feasible previously with 4xMGs or that could only be explored under BSL-4 conditions with infectious EBOV. We show the successful production of WT- and MA-EBOV trVLPs and their ability to infect target cells, to transcribe genome-encoded proteins, and, when assisted with EBOV helper proteins VP30, VP35, and L, to replicate the genome and amplify mRNA and protein production.

A system that can assess the cellular response to WT- and MA-trVLPs would be very powerful to conduct studies in relevant murine cell targets like monocytes and hepatocytes [20,31]. While there are other potential methods of gene delivery to cells, including transfection of nucleic acids, they can face cell-type-dependent challenges like poor transfection efficiency. In contrast, the 5xMG approach through trVLP infection not only enables simultaneous expression of NP and VP24 but also more accurately mimics authentic viral protein expression and cellular immune response than other potential over-expressing gene delivery methods. Although previous work identified that S72G in NP and T50I in VP24 are both required to confer virulence to MA-EBOV [21], the exact role(s) that these mutations play remain elusive. Results from previous studies point to a hypothesis that the nature of MA-EBOV pathogenesis lies in overcoming host innate immunity, demonstrated by: susceptibility of IFN-α/β receptor (IFNAR)-knockout and mitochondrial antiviral signaling protein (MAVS)-knockout mice to WT-EBOV [32,33] and a resistance of MA-EBOV to type I IFN treatment [21]. We propose that innate immunity be the target of future studies using this system.

The present study demonstrates that the T7 polymerase-driven 5xMG generates significantly higher reporter activity in transfected 293T cells compared to the pol II-driven 5xMG systems (CMVp and CAGp), using 125 ng minigenome plasmid in a 12-well plate format. Interestingly, this promoter preference appears to be reversed in the case of monocistronic EBOV minigenomes [24]; when 250 ng or less of the minigenome plasmid was used for transfection at the same scale, reporter activity was markedly higher with the CAGp system than with the T7p system. It is worth emphasizing that this is a fair comparison, as both insights were obtained in minigenome-transfected 293T (p0) cells. Thus, these insights suggest that genome length-dependent effects may influence the efficiency of minigenome expression from the plasmid. It is possible that changes in genome length affect the RNA secondary structures, potentially altering the cleavage efficiency of the ribozyme and resulting in nonfunctional minigenome termini [34]. Further investigation is needed to understand the underlying mechanisms, which could have significant implications for designing viral minigenomes and their applications.

Determining trVLPs titers is essential for conducting comparative functional analyses between WT- and MA-5xMG-trVLPs. Titration was attempted by quantifying vRNA within trVLPs by RT-qPCR. However, the resulting amplification indicated the presence of trailer and NP sequences in a control sample where no reverse transcriptase was added, despite no such amplification in the no-template control (Appendix A); this suggests the presence of a carried-over plasmid DNA template. This aberrant amplification occurred despite successive treatments of RNA with two different DNAses, implying a difficulty in accurate trVLP titration by this method. In this study, we instead utilized a reporter-based titration experiment to evaluate the infectivity of our two trVLP preparations at varying dilutions in a non-human, non-mouse cell line. This tool will allow us to have a standard for comparing WT- and MA-5xMG-trVLPs in a “neutral” system. However, we acknowledge that there may be an intrinsic bias based on the VeroE6 cell line; therefore, it would be prudent to test the bioactivity of trVLPs in a variety of cell lines to establish accurate quantification.

In summary, we demonstrate a novel, NP gene-containing EBOV polycistronic minigenome system as a useful tool. Continued investigations into the interactions among EBOV NP, VP24, and host immune responses can enhance our understanding of EBOV pathogenesis and provide insights for new strategies for therapeutic intervention.

## 4. Materials and Methods

### 4.1. Cell Culture

HEK293T/17 (ATCC CRL-11268), Huh7 (a kind gift from Dr. Yoshiharu Matsuura, Osaka University, Osaka, Japan), and VeroE6 (ATCC CRL-1586) cell lines were maintained in Dulbecco’s Modified Eagle Medium (Gibco, Waltham, MA, USA) supplemented with 10% (*v*/*v*) fetal bovine serum and 1% (*v*/*v*) penicillin and streptomycin. Cells were incubated in 5% CO_2_ at 37 °C.

### 4.2. Plasmids

EBOV (Mayinga variant) protein-expressing plasmids pCAGGS-NP, -VP35, -VP30, and -L were previously generated and described [35,36]. The initial tetracistronic minigenome plasmid was generated by assembling four synthesized gene fragments (GenScript Biotech, Piscataway, NJ, USA) into a plasmid backbone. The pentacistronic minigenome plasmid was generated by inclusion of the NP ORF into the tetracistronic minigenome plasmid through In-Fusion^®^ (Takara Bio, San Jose, CA, USA) cloning of an NP sequence-containing PCR product. The minigenome constructs driven by CAG and T7 promoters were generated by traditional restriction enzyme cloning and PCR techniques. The mouse-adaptation mutations were introduced to the pentacistronic minigenome through PCR mutagenesis of NP and VP24 fragments using iProof High Fidelity DNA Polymerase (Bio-Rad, Hercules, CA, USA) followed by In-Fusion^®^ (Takara Bio, San Jose, CA, USA) ligation. After purification, plasmid preparations were performed with endotoxin-free DNA purification kits and/or subjected to endotoxin removal. Plasmid sequences were confirmed via Sanger sequencing or next-generation sequencing before use.

### 4.3. Minigenome Transfections

For p0 transfections, HEK293T cells were seeded in 6-well plates. After 24 h, cells were transfected with a mixture comprising a 3:1 ratio of TransIT-LT1 Transfection Reagent (Mirus Bio, Madison, WI, USA) with plasmid DNA in OptiMEM media (Gibco, Waltham, MA, USA). Plasmid amounts for 4xMG: NP 125 ng, VP35 125 ng, VP30 75 ng, L 1000 ng, and minigenome 250 ng. Plasmid amounts for 5xMG: NP 75 ng, VP35 125 ng, VP30 250 ng, L 500 ng, and minigenome 125 ng. For experiments utilizing a T7 polymerase-driven minigenome, 250 ng of pCAGGS-T7pol was included in transfection reactions. To control for transfection efficiency, we included 10 ng of a pCAGGS-Luc2, which encodes a firefly luciferase gene, in each reaction.

For experiments transfecting the MA-5xMG, a pCAGGS-NP expression plasmid was used, which has the S72G mutation introduced into the NP ORF.

### 4.4. Minigenome Plasmid Titration

Helper plasmid titrations were conducted in HEK293T cells seeded in 12-well plates. Plasmid quantities were increased from 0 ng to 0.5 to 1.5 µg, depending on the plasmid size. An empty pCAGGS vector was used to account for the total plasmid deficit and compensate for changes in the total plasmid in each reaction. Cells were transfected with a mixture comprising a 3:1 ratio of TransIT-LT1 Transfection Reagent (Mirus Bio, Madison, WI, USA) with plasmid DNA in OptiMEM media (Gibco, Waltham, MA, USA). Following a 72-h incubation, whole cell lysates were harvested in 1× Passive Lysis Buffer (PLB) and used for NanoLuc Dual Reporter Assay (Promega, Madison, WI, USA).

### 4.5. Luciferase Reporter Assays

NanoLuc and NanoLuc Dual Reporter Assays (Promega, Madison, WI, USA) were performed according to the manufacturer’s instructions. Cells were lysed in 1× PLB before use. All luciferase assays were run in Costar 96-well white opaque plates using a Synergy H1 (Agilent, Santa Clara, CA, USA) or GloMax Explorer (Promega, Madison, WI, USA) plate reader.

### 4.6. trVLP Production

trVLPs were generated from minigenome transfections as described above. After 72 h, cell supernatants were harvested and clarified at 1000× *g* for 10 min. Clarified trVLP preparations were aliquoted and stored at −80 °C until use.

### 4.7. trVLP Infection of Helper-Expressing Huh7

Huh7 cells underwent reverse transfection for infection experiments. Transfection was achieved with 75 ng of pCAGGS-NP, 250 ng of -VP30, 125 ng of -VP35, 500 ng of -L, and 250 ng of -TIM-1 in OptiMEM media (Gibco, Waltham, MA, USA), then vortexed with TransIT LT-1 Transfection Reagent (Mirus Bio, Madison, WI, USA) at a 3:1 ratio. Transfection complexes were added to coat 12-well plates for 20 min. Huh7 cell suspensions were then added to the wells and incubated overnight, followed by trVLP infection the next day. Clarified trVLP-containing supernatant (400 µL) was transferred to each well for infection, incubated at 37 °C with gentle rocking of the plate in 15-min intervals. Incubations were carried out at 37 °C from 1 to 72 h before harvest in 1× PLB for downstream luciferase assay or western blot.

For serial passaging in Huh7, supernatants were harvested as described in Section 4.6. Clarified supernatants were immediately transferred to pre-transfected Huh7 for infection. Cell lysates from p1, p2, and p3 were harvested in 1× PLB for downstream NanoLuc Reporter Assays.

### 4.8. Protease Protection Assay

The trVLPs used for the protease protection assay were concentrated via ultracentrifugation. Following clarification at 1000× *g*, trVLP supernatant was loaded into an ultracentrifuge tube above a 20% sucrose cushion. Samples were spun at 32,000 rpm in a Beckman Coulter Optima L-80 XP Ultracentrifuge for 2 h before removal of the media/sucrose mixture and gentle pellet resuspension in 1× PBS. Samples were then mixed with 0.1% trypsin, 1% Triton X-100, or a combination of both before a 30-min incubation at room temperature. Samples were then prepared for Western blotting.

### 4.9. Western Blotting

Samples were mixed 1:1 with Laemmli Sample Buffer (Bio-Rad), then heated at 95 °C for 10 min before being loaded and run on a 4–15% gradient polyacrylamide SDS-PAGE gel (Mini-PROTEAN TGX Stain-Free Gel, Bio-Rad, Hercules, CA, USA). After electrophoresis, the protein was transferred using a Trans-Blot SD-Semi Dry Transfer Cell (Bio-Rad, Hercules, CA, USA) to a PVDF membrane for blotting. VP40 detection was achieved with a mouse anti-VP40 antibody (5B12, IBT Bioservices, Rockville, MD, USA) diluted at 1:2000, then an anti-mouse horseradish peroxidase (HRP) secondary antibody (Jackson ImmunoResearch, West Grove, PA, USA) diluted at 1:10,000. For NP and GP detection, a custom rabbit polyclonal antibody against the NP peptide sequence “Cys-PAVSSGKNIKRT” (Biomatik, Kitchener, ON, Canada) and rabbit anti-EBOV GP pAb (IBT Bioservices, Rockville, MD, USA) diluted 1:2000 were used, followed by secondary detection with anti-rabbit HRP-linked antibody (Cell Signaling Technologies, Danvers, MA, USA) diluted 1:10,000. β-tubulin was detected with rabbit polyclonal anti-β-tubulin antibody (Abcam, Waltham, MA, USA) diluted 1:2000 and the previously described anti-rabbit secondary. Western blots were sensitized using SuperSignal West Pico PLUS Chemiluminescent Substrate (Thermo Fisher Scientific, Waltham, MA, USA) or SuperSignal West Femto Maximum Sensitivity Substrate (Thermo Fisher Scientific, Waltham, MA, USA) and imaged on a Bio-Rad ChemiDoc.

### 4.10. RT-qPCR Assays

RNA extraction was performed by lysing trVLPs in TRIzol LS (Thermo Fisher Scientific, Waltham, MA, USA). Purification was achieved with the Direct-zol RNA MiniPrep Kit (Zymo Research, Tustin, CA, USA) with in-column DNase I treatment according to the manufacturer’s directions. vRNA quantification was conducted using 2-step RT-qPCR methods. Reverse transcription was performed using the SuperScript III or SuperScript IV Reverse Transcriptase kit (Thermo Fisher Scientific, Waltham, MA, USA) per manufacturer’s instructions with an EBOV trailer-specific primer (sequence: CTATATTTAGCCTCTCTCCC) for vRNA amplification. The resulting cDNA was subjected to qPCR with SYBR Green (Applied Biosystems, Waltham, MA, USA). Primers used: EBOV trailer fwd “GTTGCGTTAAATTCATTGCG”; EBOV trailer rev “CTATATTTAGCCTCTCTCCC”; EBOV NP fwd “GCAAGACGAGCAACAAGATC”; EBOV NP rev “CAGCATCAAATGGCCCCTGTG”. Cycling conditions are as follows: initial denaturation at 50 °C for 2 min, then 95 °C for 10 min, followed by 40 cycles of 15 sec denaturation at 95 °C and 1 min annealing/extension step. Quantification was done by comparing Ct values to a standard curve. A standard curve was generated by using dilutions of an in vitro-transcribed pentacistronic minigenome RNA species. Assays were conducted with an Applied Biosystems QuantStudio 3 and analyzed with its QuantStudio Design and Analysis software.

### 4.11. trVLP Infection of Helper-Expressing VeroE6

VeroE6 cells underwent reverse transfection for infection experiments. Transfection was achieved with 100 ng of pCAGGS-mCherry (transfection control), 125 ng of -VP30, 62.5 ng of -VP35, 250 ng of -L, and 100 ng of -TIM-1 in OptiMEM media (Gibco, Waltham, MA, USA), then vortexed with TransIT LT-1 Transfection Reagent (Mirus Bio, Madison, WI, USA) at a 3:1 ratio. Transfection complexes were added to coat 24-well plates for 20 min. Cell suspensions were then added to the wells and incubated overnight, followed by trVLP infection the next day. Clarified trVLP-containing supernatants, at the indicated dilutions, were transferred to each well for infection, incubated at 37 °C with gentle rocking of the plate in 15-min intervals. Incubations were carried out at 37 °C for 72 h before harvest 1× PLB for downstream luciferase assay.

## Figures and Tables

**Figure 1 viruses-17-00688-f001:**
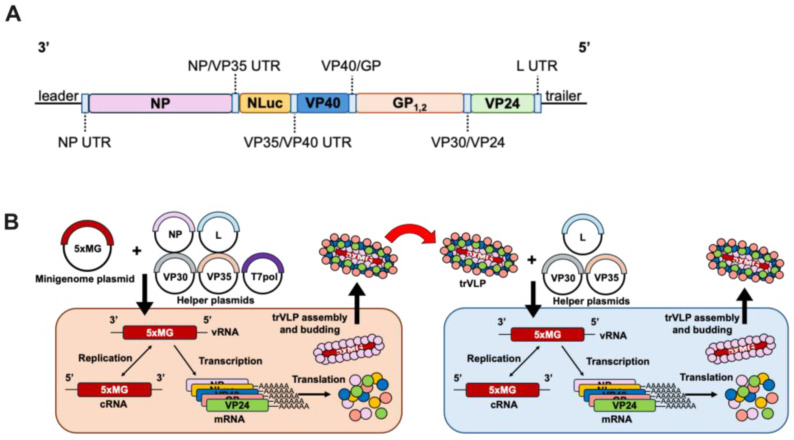
An NP Gene-containing Pentacistronic EBOV Minigenome for Modeling Infection. (**A**) Schematic of the pentacistronic minigenome (5xMG) construct organization. 3’ leader and 5’ trailer sequences are indicated at the genome termini. Viral open reading frames are indicated for NP, VP40, GP, and VP24 genes with the respective intergenic and untranslated regions (UTRs) indicated in light blue boxes. A NanoLuc (NLuc) reporter gene is included, flanked by VP35 non-coding sequences. (**B**) Schematic of 5xMG in producer and target cells. Producer cells (p0) are transfected with plasmids encoding NP, VP30, VP35, L genes, and the 5xMG. Replication and transcription of viral RNA (vRNA) can occur to generate complementary RNA (cRNA) and messenger RNA (mRNA). The proteins are then produced via translation and transcription, and replication-competent virus-like particles (trVLPs) are formed and bud from the producer cells. trVLPs can be transferred to target cells (p1) for modeling infection. Following transfection of p1 with plasmids encoding VP30, VP35, and L genes, 5xMG carried by trVLPs is transcribed and replicated as in p0. trVLPs can also be produced from p1 for further infection.

**Figure 2 viruses-17-00688-f002:**
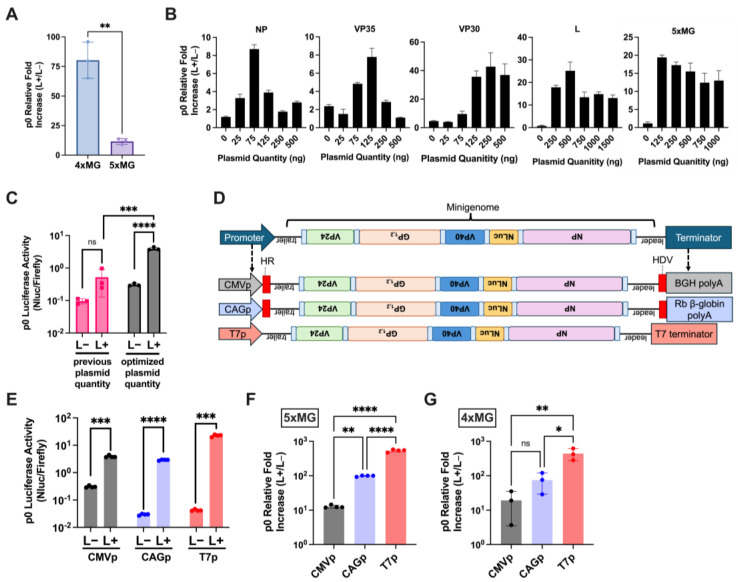
Plasmid Optimization and the T7 Bacteriophage Promoter Increase the 5xMG Efficiency in Producer Cells. (**A**) Luciferase assay of 293T cells transfected with CMVp-4xMG or -5xMG at 72 h post-transfection. Data are shown as NanoLuc luminescence of the L+ condition relative to L−. Statistical analysis was performed by unpaired, two-tailed *t*-tests. (**B**) Titration of EBOV helper expression plasmids NP, VP35, VP30, L, and the 5xMG plasmid. Plasmid quantities were increased from zero to 1000 nanograms per reaction. The unchanging plasmid quantities were: 125 ng NP, 125 ng VP35, 75 ng VP30, 1000 ng L, and 250 ng minigenome plasmid. Data are shown as NanoLuc luciferase signal of the L+ condition relative to the L− condition. Cell lysates were harvested 72 h post-transfection. (**C**) Comparison of previous and optimized plasmid quantification for 5xMG transfection in 293T cells, 72 h post-transfection. The initial transfection ratios were: 125 ng NP, 125 ng VP35, 75 ng VP30, 1000 ng L, and 250 ng minigenome plasmids in a 12-well plate format. The optimized plasmid quantities were: 75 ng NP, 125 ng VP35, 250 ng VP30, 500 ng L, and 125 ng minigenome plasmids in a 12-well plate. Data are shown as 5xMG luciferase signal relative to transfection control luciferase (NanoLuc/firefly). Conditions are either absent for the pCAGGS-L plasmid (L−) or present for the L plasmid (L+). Statistical analyses were performed by unpaired, two-tailed *t*-tests. (**D**) Schematic of 5xMG expression plasmids with the minigenome cassette in negative-sense orientation, flanked by a cytomegalovirus (CMV), chicken beta-actin (CAG), or T7 bacteriophage promoter and a termination element. The CMVp and CAGp constructs both contain a hammerhead ribozyme (HR) and a hepatitis delta virus ribozyme (HDV), indicated by red blocks, for generating precise genome ends. (**E**) Promoter selection results as a dual luciferase reporter assay 72 h post-transfection. Conditions are either absent for the pCAGGS-L plasmid (L−) or present for the L plasmid (L+). Data are shown as 5xMG luciferase signal relative to transfection control luciferase (NanoLuc/firefly). Statistical analyses were performed by unpaired, two-tailed *t*-tests. (**F**) Luciferase assay of 293T cells transfected with the 5xMG driven by either the CMV, CAG, or T7 promoters at 96 h post-transfection. Statistical analysis was performed by an ordinary one-way ANOVA with Tukey’s multiple comparisons test. (**G**) Luciferase assay of 293T cells transfected with the 4xMG driven by either the CMV, CAG, or T7 promoters at 96 h post-transfection. Data are shown as NanoLuc luminescence of the L+ condition relative to L−. Statistical analysis was performed by an ordinary one-way ANOVA with Tukey’s multiple comparisons test. For panels (**A**–**C**,**G**), data are shown with mean ± SD (n = 3 independent biological replicates). For panels E and F, data are shown with mean ± SD (n = 4 independent biological replicates). ns > 0.05, * *p* < 0.05, ** *p* ≤ 0.01, *** *p* ≤ 0.001, **** *p* ≤ 0.0001.

**Figure 3 viruses-17-00688-f003:**
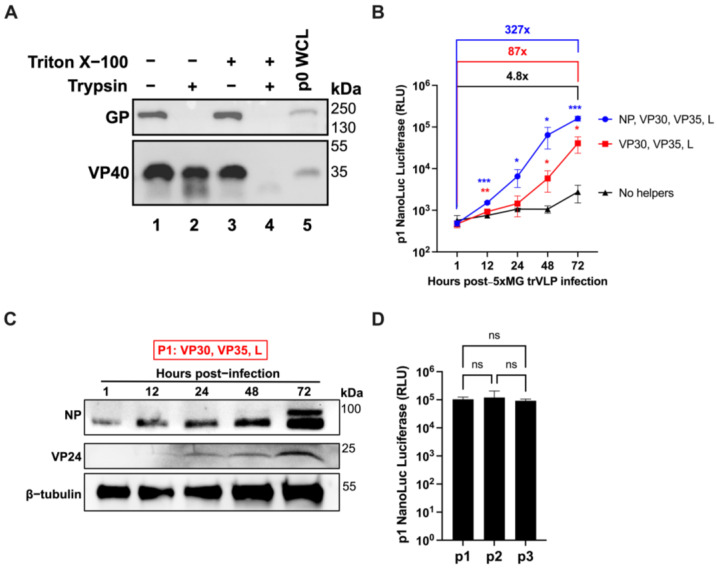
5xMG-generated trVLPs Infect and Amplify in Huh7 Target Cells. (**A**) Western blot for EBOV GP and VP40 from 5xMG trVLP fractions subjected to protease protection assay. Treatments of trVLP samples are indicated: PBS (lane 1), trypsin (lane 2), Triton X-100 (lane 3), or trypsin and Triton X-100 (lane 4). Lane 5 is whole cell lysate (WCL) from 5xMG-transfected 293T cells. (**B**) NanoLuc luciferase assay of trVLP-infected Huh7 cells. Cells were transfected with the indicated helper protein expression plasmids or an empty vector plasmid. After 24 h of transfection, cell monolayers were infected with 0.4 mL of clarified trVLP-containing supernatant. Data are shown with mean ± SD (n = 4 independent biological replicates). Statistical analysis was performed as a two-way ANOVA with repeated measures. * *p* ≤ 0.05, ** *p* ≤ 0.01, *** *p* ≤ 0.001. (**C**) Western blot for EBOV NP and VP24 in trVLP-infected Huh7. Whole cell lysates were derived from the ‘VP30, VP35, and L’ samples shown in panel B at the indicated timepoints. (**D**) NanoLuc luciferase assay of trVLP-infected Huh7 cells after serial passaging. Cells were pre-transfected with helper plasmids prior to infection. trVLP-containing supernatants were harvested at 72 h post-infection and transferred to new cells for passaging. Data are shown with mean ± SD (n = 3 independent biological replicates). Statistical analysis was performed as a one-way ANOVA. ns *p* > 0.05.

**Figure 4 viruses-17-00688-f004:**
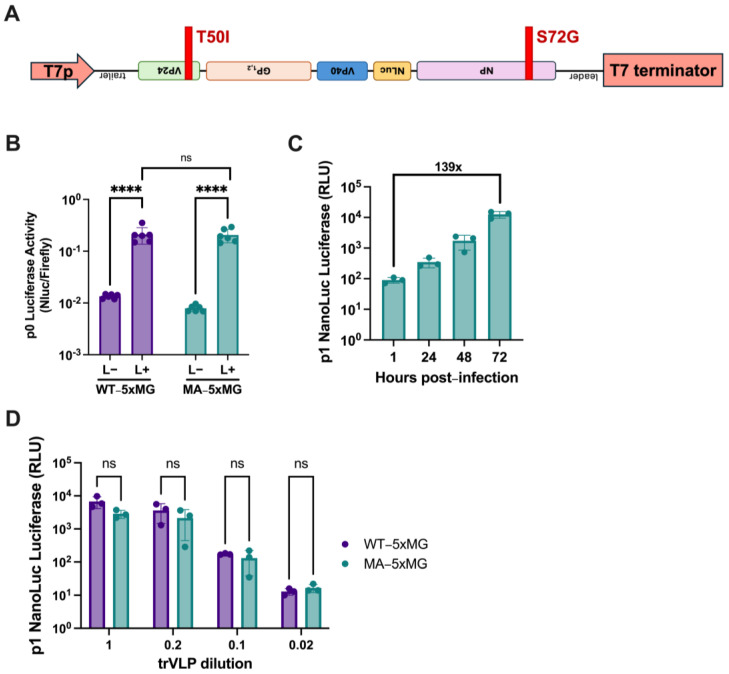
Introduction of Mouse-Adaptation Mutations Does Not Significantly Impact 5xMG Efficiency. (**A**) Diagram of the T7p-driven mouse-adapted 5xMG (MA-5xMG). MA mutations were introduced in the NP and VP24 genes at the indicated positions. (**B**) Dual luciferase reporter assay results of 293T cells transfected with WT- or MA-5xMG at 72 h post-transfection. Data are shown as 5xMG luciferase signal relative to transfection control luciferase (NanoLuc/firefly). Conditions are either absent of the pCAGGS-L plasmid (L−) or present for the L plasmid (L+). Statistical analyses were performed by unpaired, two-tailed *t*-tests. Data are shown with mean ± SD (n = 6 independent biological replicates). (**C**) NanoLuc luciferase assay of MA-trVLP-infected Huh7 cells. Cells were pre-transfected with helper plasmids prior to infection. Cell lysates were harvested at the indicated time points. Data are shown with mean ± SD (n = 3 independent biological replicates). (**D**) NanoLuc luciferase assay of helper-expressing VeroE6 cells infected with indicated dilutions of trVLP-inoculum. Cell lysates were harvested for reporter assay at 72 h post-infection. Data are shown with mean ± SD (n = 3 independent biological replicates). For B, statistics were performed as unpaired, two-tailed *t* tests. For D, statistics were performed as multiple unpaired *t* tests. ns *p* > 0.05, **** *p* ≤ 0.0001.

## Data Availability

The original data presented in the study are available from the corresponding authors upon request.

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
