# Peer review of "Development of a Pentacistronic Ebola Virus Minigenome System"

_viruses, 2025, doi:10.3390/v17050688_

Round 1
Reviewer 1 Report (Previous Reviewer 2)
Comments and Suggestions for Authors
The authors have updated the manuscript, addressing all of the concerns I had with the originally-submitted version. In my opinion, it is significantly improved and ready for publication.
Author Response
Comment 1: The authors have updated the manuscript, addressing all of the concerns I had with the originally-submitted version. In my opinion, it is significantly improved and ready for publication.
Response 1: Thank you!
Reviewer 2 Report (Previous Reviewer 3)
Comments and Suggestions for Authors
This is a revised version of a previously reviewed manuscript. The authors have removed parts of the manuscript comparing the cellular responses of RA264.7 cells after infection with trVLPs encoding either WT or MA NP. This mostly addresses my previous concerns that the trVLPs they used for these experiments indeed contain a mix of WT and MA NP. As such it would no longer be necessary to discuss these limitations in the discussion, as the experiments the limitations apply to no longer are part of the manuscript.
However, the same criticism still applies to figure 4 and section 3.5. Again, the authors cannot draw any clear conclusions regarding an effect (or the absence of an effect) of the mouse adaptations on viral RNA synthesis, as they appear to use wildtype NP helper plasmids for these assays (at least they do not indicate otherwise). I would suggest to remove these parts of the manuscript, as they are, in my view, potentially misleading and do not really add anything to the overall conclusions of the manuscript.
That being said, the establishment of the pentacistronic minigenome itself, and particularly the experiments with infecting p1 cells lacking NP are intriguing, and justification enough for publishing the manuscript.
Author Response
Comment 1: This is a revised version of a previously reviewed manuscript. The authors have removed parts of the manuscript comparing the cellular responses of RA264.7 cells after infection with trVLPs encoding either WT or MA NP. This mostly addresses my previous concerns that the trVLPs they used for these experiments indeed contain a mix of WT and MA NP. As such it would no longer be necessary to discuss these limitations in the discussion, as the experiments the limitations apply to no longer are part of the manuscript.
However, the same criticism still applies to figure 4 and section 3.5. Again, the authors cannot draw any clear conclusions regarding an effect (or the absence of an effect) of the mouse adaptations on viral RNA synthesis, as they appear to use wildtype NP helper plasmids for these assays (at least they do not indicate otherwise). I would suggest to remove these parts of the manuscript, as they are, in my view, potentially misleading and do not really add anything to the overall conclusions of the manuscript.
Response 1: Thank you for this comment. To address this, we have provided alterations to Figure 4. We have repeated these experiments by utilizing the appropriate NP helper plasmids for transfection (i.e. WT-NP for WT-5xMG and MA-NP for MA-5xMG). Panel B shows this as the p0 transfection of 293T cells. Panels C and D show infection of these trVLPs in Huh7 and VeroE6 cells. These updated Results are described in lines 237-274.
Comment 2: That being said, the establishment of the pentacistronic minigenome itself, and particularly the experiments with infecting p1 cells lacking NP are intriguing, and justification enough for publishing the manuscript.
Response 2: Thank you!
Reviewer 3 Report (New Reviewer)
Comments and Suggestions for Authors
Zell and colleagues engineered a pentacistronic EBOV minigenome (5xMG) system by adding the nucleoprotein (NP) gene to an existing tetracistronic design. The functionality of the 5xMG was demonstrated and optimized using different promoters and helper plasmid ratios. NP expression in passage 1 with and without NP helper plasmid added in trans suggested bonafide trVLP presence. The Author’s also introduced mouse-adapted EBOV nonsynonymous changes to the NP and VP24 but did not observe any phenotypical differences in passage 1 infected cells.
Overall, the manuscript is polished and well-structured with nice figures and clear writing. Nevertheless, while the 5xMG is novel and may have useful applications, the Authors could benefit from additional experiments and analysis to support their main hypothesis. The Authors prove that the assay works and that all components are expressed in vitro. The study design included optimization efforts in choosing the best promoter and helper plasmid ratios, which are foundational to the development of any MG system.
However, the evidence for the 5xMG’s greater biological applications should be expanded upon. Specifically, the Author’s should demonstrate how the inclusion of NP into the tetracistronic MG can produce biologically relevant data and bring value to existing biologically contained EBOV research. Please see the following comments:
- The Author’s state in their penultimate introductory sentence (lines 64-65) that they wanted to “create a system to evaluate the effects of mutations found in MA-EBOV…” by introducing “mutations into the NP and VP24 genes within the 5xMG…”. First, as modeled in a preceding work by Watt et al., 2014 (e.g. Fig 3B), the stability across multiple generations needs to be demonstrated.
- The Author’s include a caveat in their discussion (lines 284-287) mentioning how the NP-helper plasmid is likely carrying over into p1, possibly mixing WT and MA-EBOV NP populations, which could affect downstream results. Additionally, they also mention in the same paragraph that they could have included a NP-helper that includes the NP point mutation starting from P0 to match the MA-5xMG mutation. Implementing matching NP sequences for trVLP production is feasible with recent synthetic biology developments and would be essential for testing any future NP mutations of interest.
- The VeroE6 comparison in Figure 4C could serve as a starting point for answering whether the MA-5xMG-trVLPs do have a phenotypical difference compared to the WT version. Figure 4 of Ebihara et al., 2006, demonstrated the IFN-treatment impact on MA-EBOV infected RAW 264.7 cells. Adapting/mirroring this experimental design to test the novel MA-5xMG-trVLP would certainly provide a strong justification for its additional use and further exemplify the value of the 5xMG-trVLP tool for laboratories restricted to BSL2 spaces.
Additional comments:
Line 17: trVLPs should be written out here, as this is the first time the abbreviation appears.
Figures 2C (line 161) and 4B: Please include direct statistical comparisons between L+ columns.
Figure 4B & C: B used 293T cells and C used VeroE6 cells. Were there any trVLP comparisons done using any mouse specific cell lines?
Author Response
Comment 1: The Author’s state in their penultimate introductory sentence (lines 64-65) that they wanted to “create a system to evaluate the effects of mutations found in MA-EBOV…” by introducing “mutations into the NP and VP24 genes within the 5xMG…”. First, as modeled in a preceding work by Watt et al., 2014 (e.g. Fig 3B), the stability across multiple generations needs to be demonstrated.
Response 1: Thank you for this suggestion! To address this, we performed 3 passages of the 5xMG trVLPs in helper plasmid-transfected human Huh7 cells and have included these results as Figure 3D which are described in the text from lines 230-232.
Comment 2: The Author’s include a caveat in their discussion (lines 284-287) mentioning how the NP-helper plasmid is likely carrying over into p1, possibly mixing WT and MA-EBOV NP populations, which could affect downstream results. Additionally, they also mention in the same paragraph that they could have included a NP-helper that includes the NP point mutation starting from P0 to match the MA-5xMG mutation. Implementing matching NP sequences for trVLP production is feasible with recent synthetic biology developments and would be essential for testing any future NP mutations of interest.
Response 2: Thank you for this comment. We agree this is important and feasible. We have since repeated the experiments utilizing MA-5xMG and have included the appropriate NP sequence in p0 (i.e. WT-NP for WT-5xMG and MA-NP for MA-5xMG). This is reflected in an updated Figure 4 and is described in the text from lines 237-274.
Comment 3: The VeroE6 comparison in Figure 4C could serve as a starting point for answering whether the MA-5xMG-trVLPs do have a phenotypical difference compared to the WT version. Figure 4 of Ebihara et al., 2006, demonstrated the IFN-treatment impact on MA-EBOV infected RAW 264.7 cells. Adapting/mirroring this experimental design to test the novel MA-5xMG-trVLP would certainly provide a strong justification for its additional use and further exemplify the value of the 5xMG-trVLP tool for laboratories restricted to BSL2 spaces.
Response 3: Thank you; we fully agree with this comment. We have included an updated comparison of WT- and MA-trVLP infection in VeroE6 cells as Figure 4D. This manuscript is a resubmission and in the previous version we included experiments involving infection of RAW264.7 macrophages and downstream analysis of that infection. Following reviewer critiques, we determined 2 issues within our experimental design and results: 1) differential level infection with WT- and MA-trVLPs despite titration of trVLP batches and 2) reporter signal observed in the infected cells without helper protein pre-transfection appeared to be result of pseudo-transduction, and not authentic minigenome expression. Due to these difficulties, we removed these experiments from the manuscript.
Unfortunately, RAW264.7 mouse cells are very difficult to transfect with one plasmid, let alone the multiple plasmids needed as helpers to replicate and amplify the trVLPs. Transfection efficiencies with 1 plasmid are on the order of 1-3% GFP+ with Transit LT, Lipofectamine 3000, or Fugene 6.
For the helper plasmids to amplify trVLP signal and EBOV protein expression from the minigenome, all helper plasmids must co-transfect the same cell. If one plasmid only transfects 1-3% of RAW cells, this makes it likely that only 1-3% of the cells get all of the plasmids at best (assuming that all were co-packaged and succeeded by the same mechanism). If each plasmid stochastically transfects cells independently, in a worst case scenario, the odds might be vanishingly small (ie. 7 plasmids at probability of 0.01 = ~ 0.017 or ~ 10-14.
For this reason, and the pseudo-transduction problem, we do not believe we can at this time perform the experimental mimic of the Ebihara in RAW cells.
Comment 4: Line 17: trVLPs should be written out here, as this is the first time the abbreviation appears.
Response 4: We have since changed the text in lines 17-18 to fix this.
Comment 5: Figures 2C (line 161) and 4B: Please include direct statistical comparisons between L+ columns.
Response 5: We have added the statistical comparison for the two L+ columns in Figure 4B.
Comment 6: Figure 4B & C: B used 293T cells and C used VeroE6 cells. Were there any trVLP comparisons done using any mouse specific cell lines?
Response 6: See above comment about prior experiments with mouse cell lines.
Round 2
Reviewer 3 Report (New Reviewer)
Comments and Suggestions for Authors
No further changes are suggested.
This manuscript is a resubmission of an earlier submission. The following is a list of the peer review reports and author responses from that submission.
Round 1
Reviewer 1 Report
Comments and Suggestions for Authors
Zell and colleagues constructed and characterized a pentacistronic Ebola virus (EBOV) minigenome. Furthermore, they employed trVLPs, based on this system, to determine the impact of mouse adaptation (MA, i.e. mutations in NP and VP24) on trVLP-induced alterations in gene expression in mouse macrophages. They show that MA trVLP induce less innate/inflammatory responses as compared to WT trVLPs. These findings are of interest to the field. However, several points remain to be addressed.
Major
The differential impact of WT versus MA trVLP on gene expression is appreciated. However, more information is needed for solid conclusions: Did RNAseq confirm the presence of comparable amounts of WT and trVLP RNAs in the infected cells? Does IFN treatment of target macrophages have a more pronounced effect on WT as compared to MA trVLP infection? Do MA NP and VP24 but not their WT counterparts reduce induction of the IFN promoter by RLR?
Figure 4D indicates that roughly 3000 RLU were measured in trVLP infected RAW macrophages as compared to 10 RLU in mock treated cells. It is important to show that this difference is due to infectious entry, using either a non-functional GP mutant or a neutralizing monoclonal antibody.
Minor
“which used 125 ng NP, 125 ng VP35, 75 ng VP30, 1,000 ng 155 L, and 250 ng minigenome plasmids.” This statement is only fully useful if information on the cell culture dish is provided (for instance “per well of a 12-well plate”).
“However, the 5xMG construct exhibited an approximately 10-fold lower reporter activity compared to 4xMG (Figure 2A).” This statement cannot be verified by the reader since unprocessed luciferase counts are not shown. Furthermore, it should be stated that 5 x MG responded less robust to the presence of polymerase (and this response should be independent from overall luciferase activity).
Abstract: „causes hemorrhagic disease“. This is not essential but the authors might want to take into account that Ebola hemorrhagic fever was renamed Ebola virus disease because hemorrhages do not occur in all patients and are not responsible for a lethal outcome.
Please revise the manuscript for minor grammar and style issues like “EBOV is required be handled in” and “extreme safety concerns” (extreme may not be a good word to describe safety concerns).
Reviewer 2 Report
Comments and Suggestions for Authors
This manuscript describes the generation and use of a novel pentacistronic trVLP system that facilitates the study of NP mutations within the context of a BSL-2 compatible trVLP system. The authors use this system to study the effects of adaptations observed in mouse-adapted Ebola virus on infection of the mouse macrophage RAW264.7 cell line, providing an interesting proof-of-principle use for their system. This is a well put together manuscript that, with some minor additions/edits, would be a welcome addition to the field.
Concerns:
11. Figures 2E-2G: The authors describe the T7 system as superior to the polII systems (CMV and CAG promoters) but only show one time point. How do these systems compare at 2 or 4 dpi? Also, the authors only compared these systems in 293T cells. It might be worthwhile to discuss that for trVLP systems it is most critical to compare the efficiency in these P0 cells, as the MG will be transmitted via the trVLPs for any subsequent passages (P1+), whereas for monocistronic MG assays, it is more important to compare them in the cell type in which they will be used, e.g. as in Nelson et al 2018.
22. Figure 3B: The authors should describe why they observed a 3-4 fold increase in trVLP activity when they pre-transfected their P1 cells with NP versus without NP. Also, this is presumably the 5xMG (as in the figure legend) but it wouldn’t hurt to add that to the figure.
33. Figure 5: The authors do not describe the infection rate of the RAW264.7 cells in this experiment. Although the authors show similar levels of infection of these cells previously (Fig. 4D), that was a different experiment. This could very easily be rectified by quantifying & presenting the viral reads observed in their RNA-seq dataset. The comparison of a single trVLP transcript’s reads (e.g. NP) would be fine.
Minor concerns:
11. Figure 1A legend: While untranslated regions are non-coding they are, by definition, not “intergenic”. This should be corrected.
22. Line 105: Add that the cells have to be “previously” transfected in order to amplify the 5xMG delivered by trVLP.
33. Lines 109-111: It is not clear why a 5xMG would allow you to study aspects of NP biology that you couldn’t accomplish with a 4xMG using plasmids encoding different versions of NP in the P0 or P1 cells.
44. Figure 2A: “Relative to L-“ is not clear. Is this L+/L- or (L+)-(L-)? This should be clear in the y-axis label of the graph.
55. Figure 2B: While stated in the text (Lines 155-156), the amount of the other plasmids used should somehow be indicated in the figure (e.g. who much VP35, VP30, L, and 5xMG was used in the first graph?).
66. Figure 2C: Same as comment 5; the plasmid amounts should be somehow shown in the figure as this is critical for understanding the differences in the samples.
77. Figure 2D: “leader”, “trailer”, and the gene names are a little difficult to read (the leader and trailer are especially illegible). If these could be made bigger it would be helpful. Also, if the VP40 gene could be made a lighter shade of blue, it would help.
88. Figures 2F and 2G: Presumably these data are from P0 cells, but that isn’t indicated in either the figure or legend (as is done for Figure 2E).
99. Lines 178-179: Add Supplementary Figure 1 here, as the 4xMG trVLP system hasn’t been described before this.
110. Figure 3C: It should be indicated somewhere in the figure that these data are derived from P1 cells.
111. Figure 4C: Are these titers from supernatants of P0-transfected cells or P1+-transfected cells?
112. Lines 257-258: It is confusing to refer to the production of non-self-replicating trVLPs as “rescue”. It would be more accurate to say that there were similar levels of WT- and MA-trVLP production.
113. Figure 5A: Presumably the trVLPs used to infect the RAW264.7 cells were produced in 293T cells. This should be indicated in the figure legend.
Reviewer 3 Report
Comments and Suggestions for Authors
The manuscript „Development of a Pentacistronic Ebola Virus Minigenome and its Application to Model Mouse-Adapted Ebola Virus” by Zell et al. describes the generation and optimization of a pentacistronic (5cis) transcription and replication-competent virus-like particle (trVLP) system for Ebola virus (EBOV) and its subsequent use to try to compare the cellular responses of RA264.7 cells to infection with trVLPs encoding either wild-type (WT) EBOV proteins, or mouse-adapted (MA) variants of VP24 and NP.
The manuscript is well written, and particularly the establishment and optimization of the pentacistronic trVLP system is very thoroughly done and presented well. However, there are some significant concerns regarding the last part of the manuscript, i.e. the comparison of cellular responses to infection with different trVLPs. Specifically, while the authors claim that these trVLPs model either WT or MA EBOV, this is not entirely true, since in both cases (at least judging by the information provided in the manuscript) p0 cells were transfected with helper plasmids encoding WT NP. This means that the trVLPs produced in presence of the MA 5cis minigenome will contain a mixture of MA NP (derived from the minigenome) and WT NP (derived from helper plasmids). Therefore, it is very difficult to draw firm conclusions regarding an effect of the MA mutations. This problem could have been avoided if the authors had matched the genotype of the NP helper plasmid to that of the minigenome in their experiments, but at least judging by the information provided this was not the case. Alternatively, they could have used p1 supernatants (with p1 cells being pretransfected to express VP35, VP30 and L, but no NP) for infection of p2 target cells.
A second issue is that the authors only check that they have comparable amounts of infectious trVLPs in their trVLP preps derived from WT or MA 5cis minigenomes. Given that they analyze host cell responses relatively early after infection (i.e. at 24 hrs), and don’t have robust viral RNA synthesis in p1 cells, but only limited amounts of primary transcription, it is possible (if not even likely) that the responses they measure are largely dependent on sensing of VLPs at the cell surface, or during the uptake process. This sensing would detect not only infectious trVLPs, but also most likely non-infectious VLPs; however, no information is provided if the overall particle amounts in the trVLP preps from different origins are comparable. This question could have been addressed by Western Blotting of trVLP-containing supernatants from p0 cells.
These two issues need to be addressed either experimentally, or at least very clearly as limitations of the study in the discussion section. Further, the authors should consider weakening their conclusions regarding the different host cell responses.
The following specific points should be addressed:
Major points:
1) The two major shortcomings of this study as already described above, i.e. the fact that MA-trVLP preps contain a mixture of WT and MA NP, and 2) that MA- and WT-trVLP preps are not controlled for the overall VLP content, need to be clearly disclosed and discussed, and the conclusions regarding host cell responses should be weakened.
Minor points:
2) Line 64: The phrase “but only when” here is somewhat confusing. I would suggest to change it to something like “and sufficient for a virulent phenotype only when”.
3) Figure 2A and following: Specify for the Y axis that this is fold increase compared to -L. I would suggest “fold increase relative to -L”.
4) Figure 2F and G: IT is impossible to see how well the CMV-driven 5cis mg performs in these figures. I would suggest using a logarithmic scale here.
5) Line 273: Please specify that naive target cells were used.